# Improved Algorithms for Fair Matroid Submodular Maximization

**Sepideh Mahabadi**
Microsoft Research
smahabadi@microsoft.com

**Sherry Sarkar**[*]
Carnegie Mellon University
sherrys@andrew.cmu.edu

**Jakub Tarnawski**
Microsoft Research
jakub.tarnawski@microsoft.com

## Abstract

Submodular maximization subject to matroid constraints is a central problem with many applications in machine learning. As algorithms are increasingly used in decision-making over datapoints with sensitive attributes such as gender or race, it is becoming crucial to enforce fairness to avoid bias and discrimination. Recent work has addressed the challenge of developing efficient approximation algorithms for fair matroid submodular maximization. However, the best algorithms known so far are only guaranteed to satisfy a relaxed version of the fairness constraints that loses a factor 2, i.e., the problem may ask for $\ell$ elements with a given attribute, but the algorithm is only guaranteed to find $\lfloor \ell/2 \rfloor$. In particular, there is no provable guarantee when $\ell = 1$, which corresponds to a key special case of perfect matching.

In this work, we achieve a new trade-off via an algorithm that gets arbitrarily close to full fairness. Namely, for any constant $\varepsilon > 0$, we give a constant-factor approximation to fair monotone matroid submodular maximization that in expectation loses only a factor $(1 - \varepsilon)$ in the lower-bound fairness constraint. Our empirical evaluation on a standard suite of real-world datasets – clustering, recommendation, and coverage tasks – demonstrates the practical effectiveness of our methods.

The code for the paper is available at `https://github.com/dj3500/fair-matroid-submodular-neurips2025`.

## 1   Introduction

Machine learning is increasingly deployed in high-stakes decision-making, raising concerns about the propagation of bias and unfairness in automated systems. These challenges are especially acute in domains such as education, law enforcement, hiring, and credit [MMD16; Whi22; Eur22]. In response, a growing body of research has focused on developing algorithms that incorporate fairness constraints for core problems including clustering [CKLV17], data summarization [CKSDKV18], classification [ZVGG17], voting [CHV18], and ranking [CSV18].

This paper studies fairness in the context of monotone submodular maximization subject to matroid constraints. Submodular functions, which capture the principle of diminishing returns, are fundamental to a range of machine learning applications such as recommender systems [EG11], feature selection [DK11], active learning [GK11], and data summarization [LB11]. Matroids provide a general framework for modeling independence constraints, encompassing cardinality, partition, graph connectivity, and linear independence constraints.

---

[*]Part of this work was done while an intern at Microsoft Research.

39th Conference on Neural Information Processing Systems (NeurIPS 2025).

While numerous fairness definitions have been proposed, we adopt a widely used group fairness model, which partitions the universe into *disjoint* groups and enforces *lower and upper* bounds on the representation of each sensitive group in the selected set. See Section 2.1 for a precise definition. This model generalizes several fairness notions, such as proportional representation [Mon95; BLS17], diversity constraints [CCRL13; Bid06], and statistical parity [DHPRZ12]. It has been used for both submodular maximization [CSV18; CHV18; EMNTT20; EFNTT23; WFM21; TY23; YT23; ETNV24] as well as a multitude of other optimization problems, such as clustering [CKLV17; KAM19; JNN20; HMV23], voting [CHV18], data summarization [CKSDKV18], matching [CKLV19] or ranking [CHV18].

In the absence of fairness constraints, monotone submodular maximization under a single matroid constraint is very well understood, as a tight $(1-1/e) \approx 0.63$-approximation is achievable [CCPV11; Fei98]. The intersection of two matroid constraints (which we refer to as "matroid intersection") admits an almost $0.5$-approximation [LSV10]. The fair variant has been primarily explored under cardinality constraints [CHV18], where a tight $(1 - 1/e)$-approximation is also known. In the (single-pass) streaming setting, there is a $0.3178$-approximation [FLNSZ22] for the non-fair matroid version; furthermore, since the intersection of cardinality constraint and fairness can be reduced to a single matroid constraint [EMNTT20], the same approximation factor can be obtained for it.

However, the intersection of a matroid constraint and a fairness constraint seems significantly more challenging, and is still poorly understood despite two recent works devoted to studying this problem in the streaming [EFNTT23] and the classic offline [ETNV24] settings; our focus is on the latter. Following [EFNTT23], we refer to the problem as Fair Matroid Monotone Submodular Maximization (**FMMSM**). To appreciate its difficulty, consider a key special case, Monotone Submodular Perfect Matching (**MSPM**), i.e., maximizing a monotone submodular function over the collection of all *perfect* matchings in a *bipartite* graph $(V_G, E_G)$.[2] This collection of feasible sets is not downward-closed, which invalidates known algorithmic approaches.[3] The best known approximation factor for MSPM is a trivial $O(|V_G|)$-approximation; one can also apply the framework of [GHIM09] to obtain an $\widetilde{O}(\sqrt{|E_G|})$-approximation, which is superior for sparse graphs. In fact, this could possibly even be tight, as it almost matches a surprising negative result of [ETNV24] who showed a family of sparse graphs where the standard *multilinear relaxation* (commonly used in relax-and-round approaches for submodular optimization) has an integrality gap of $\Omega(\sqrt{|E_G|})$. The existence of a constant-factor approximation to MSPM was posed by [ETNV24] as an exciting open problem.

The algorithms given in [EFNTT23; ETNV24] for FMMSM circumvent the difficulty posed by the lower bound constraints by relaxing them. They obtain the following two results:

**Theorem 1.1 (Two-pass algorithm of [EFNTT23])** *There is a polynomial-time algorithm for FMMSM that violates lower bound constraints by a factor 2 and obtains $\alpha/2$-approximation, where $\alpha$ is the approximation ratio of an algorithm for maximizing a monotone submodular function under a matroid intersection constraint.*

We can have $\alpha$ be almost $1/2$ [LSV10] and thus get an almost $1/4$-approximation. ([EFNTT23] work in the streaming setting and instead use the streaming algorithm for matroid intersection of [GJS21]; this results in a $1/11.66$-approximation in two passes.) Here, violating lower bound constraints by a factor 2 means that, if a color has a lower bound of $\ell$, the solution is guaranteed to have at least $\lfloor \ell/2 \rfloor$ elements of that color. Note that in MSPM we have $\ell = 1$ and thus $\lfloor \ell/2 \rfloor = 0$.

**Theorem 1.2 ([ETNV24])** *There is a polynomial-time algorithm for FMMSM that satisfies* lower and upper *bound constraints in expectation rather than exactly, and obtains a $(1-1/e)$-approximation in expectation.*

---

[2]To see why MSPM is a special case of FMMSM, set $E_G$ as the universe, consider a partition matroid that encodes that every vertex on the left shall have degree at most 1 in the solution, and set fairness constraints so that every vertex on the right shall have degree at least 1 and at most 1.

[3]Of course, a proper subset of a perfect matching is not a perfect matching. But more importantly, the collection of all *subsets of perfect matchings* (which is downward-closed) does not belong to any of the families that are known to make approximate submodular maximization tractable. In particular, it is not a matroid, an intersection of few matroids, or a so-called $p$-extendible set system or a $p$-system [CCPV11] for $p = O(1)$.

Theorem 1.2 also guarantees certain two-sided tail bounds on the violation of each fairness constraint which apply if $\ell$ is large enough. It is the only algorithm considered in this paper that violates the *upper* bounds. The algorithm proceeds by solving and rounding the multilinear relaxation.

If we consider a relaxed version of MSPM where instead of a *perfect* matching we want a *large* matching that also has high submodular function value, then a simple greedy algorithm will yield a $1/3$-approximation (Theorem 2.5) and construct a maximal matching, thus getting $1/2$ of the maximum possible size. The results in Theorems 1.1 and 1.2 give no improvement upon this. While one can try to generalize this simple approach to FMMSM, it faces another issue that is salient in the context of fairness motivations: while at least half of the total lower bound mass will be satisfied, there could be "unlucky" colors (marginalized groups) that never get represented in the solution; this is precisely the reason why we seek fair algorithms in the first place.

## 1.1 Our contributions

In this work we provide an algorithm that satisfies the fairness constraints within a factor better than 2, while also giving guarantees for every individual group (rather than only in aggregate like the simple greedy strategy discussed above). To achieve the former, we trade off part of the objective value; to achieve the latter, we employ randomization.

**Theorem 1.3 (informal version of Theorem 3.4)** *For every $\varepsilon \in (0, 1)$ there is a polynomial-time algorithm for FMMSM whose output*

- *satisfies the matroid constraint,*
- *satisfies fairness upper bound constraints,*
- *for a group with fairness lower bound $\ell$, has in expectation at least $(1 - \varepsilon)\ell$ elements from that group,*
- *has expected size at least $(1 - \varepsilon)$ times the maximum size of any feasible solution,*
- *satisfies Chernoff-style high-probability bounds on size, as well as total fairness violation,*
- *has expected submodular function value at least $0.499 \cdot \varepsilon \cdot \mathrm{OPT}$.*

Our bound on the submodular function value is actually shown with respect to a more powerful optimum, namely, an optimal set that satisfies the matroid and upper-bound constraints, but not necessarily the lower-bound constraints. If one wants to compare to this optimum, then the $O(\varepsilon)$ factor loss in value is unavoidable. To see this, consider MSPM in a graph $P_3 \times N$ consisting of a disjoint union of $N$ paths of length 3, with a linear objective function assigning values $0, 1, 0$ to each path's edges. A perfect matching of size $2N$ has $0$ value, and a maximal matching of size $N$ has value $N$; one can interpolate between these smoothly.

We note that by instantiating $\varepsilon = 1/2$ we obtain an almost $1/4$-approximation while violating lower bounds by a factor 2, which is similar to the bounds of Theorem 1.1 ([EFNTT23]).

As a second contribution, we also employ our techniques to obtain a deterministic algorithm. There are several variants that we could formulate; we choose to show a general setting of matroid intersection, where the trade-off is between size and objective value. The relation to fairness is that an algorithm that finds a solution of maximum size that is an $\alpha$-approximation to the objective value would imply an $\alpha$-approximation algorithm for FMMSM (see [EFNTT23], Proposition C.6).

**Theorem 1.4** *For every $\varepsilon \in (0, 1)$ there is a deterministic polynomial-time algorithm for the problem of maximizing a monotone submodular function subject to two matroid constraints whose output has size at least $(1 - \varepsilon)$ times the maximum size of any feasible solution minus one, and obtains a $(0.499 \cdot \varepsilon)$-approximation to the submodular function value.*

**Experimental results.** We show the effectiveness of our algorithm empirically against prior work and natural baselines on a suite of standard benchmarks. We measure the submodular objective value and total fairness violation. Our algorithms produce solutions whose value is competitive with the highest-value baseline, which completely ignores the lower bound constraints and accordingly has the highest fairness violations. In two out of three scenarios, our algorithms dominate prior work [EFNTT23]. Finally, a key strength of our approach is the flexibility given by $\varepsilon$, allowing users to tune the balance between utility and fairness.

**Our techniques.** Let us begin with the simple setting of perfect matchings (MSPM). Consider the symmetric difference of a high-value matching $Y$ and a perfect matching $P$. This decomposes into a collection of vertex-disjoint alternating cycles and augmenting paths.

One possible algorithm is to ignore the cycles, and choose some of the augmenting paths to apply to $Y$, so that its size grows to at least $(1 - \varepsilon)|P|$. We can do this by computing the marginal contribution of the elements that $Y$ would lose in each path, and taking the least damaging paths; by submodularity, this loses at most a $(1 - \varepsilon)$ fraction of value in $Y$.

While this does ensure a large matching, some $\varepsilon$ fraction of vertices can still be "unlucky" and end up unmatched. Deterministically this would be hard to avoid (short of solving MSPM/FMMSM completely, with no fairness violation); our next idea is to choose the paths randomly in the above solution. This will work for MSPM, as long as we take care to select a $(1 - \varepsilon)$ fraction of the $|P| - |Y|$ many augmenting paths, even if we already have $|Y| \geq (1 - \varepsilon)|P|$. Then every vertex that was not matched in $Y$ has a $(1 - \varepsilon)$ probability of being matched in the new solution.

However, there are two main challenges when trying to generalize the above approach to matroid and fairness constraints. Firstly, having fairness bounds with $\ell_c < u_c$ means that $Y$ can have fewer elements than $P$ in some colors but more elements in other colors, and can even have $|Y| = |P|$ while still violating many fairness lower bounds. This means that we need to find and apply not only augmenting paths, but also alternating paths that exchange an element of an oversaturated color for one of an undersaturated color, without increasing the solution size. We show that as long as the total fairness violation is large, there are many such disjoint paths, which implies that applying a random fraction of them still retains enough value.

The second, larger obstacle arises due to dealing with general matroids. We are able to use tools from matroid theory to show the existence of many disjoint alternating or augmenting paths in an appropriate matroid intersection exchange graph whose vertices correspond to elements of $Y$ and $P$ (which were edges in the case of MSPM). We need to carefully refine the paths via an asymmetric shortcutting process to ensure that applying them leaves the solution independent in the matroid while also not disrupting the counts of elements in the colors not being exchanged. Moreover, in general, multiple augmenting paths in matroids cannot be applied simultaneously. We deal with this using an iterative framework where we apply a single path, rebuild the exchange graph, and find a new large collection of disjoint paths; we then bound the loss in value after each step.

**Paper organization.** In Section 2 we introduce all necessary notation, definitions, and useful facts. In Section 3 we describe our algorithms and prove their properties. Section 4 is devoted to the experimental evaluation. We conclude and discuss the limitations and broader impact of our work in Section 5. Additional related work is discussed in Section 1.2 in the full version (in the supplementary materials), which also contains all omitted content and skipped proofs.

## 2 Preliminaries

We denote the symmetric difference $(X \setminus Y) \cup (Y \setminus X)$ of two sets $X$ and $Y$ by $X \triangle Y$.

**Submodular functions.** We consider functions $f : 2^V \to \mathbb{R}_+$ defined on a ground set $V$. We say that $f$ is *submodular* if $f(Y \cup \{e\}) - f(Y) \geq f(X \cup \{e\}) - f(X)$ for any two sets $Y \subseteq X \subseteq V$ and any element $e \in V \setminus X$. Moreover, $f$ is *monotone* if $f(Y) \leq f(X)$ for any two sets $Y \subseteq X \subseteq V$. We assume that $f$ is given as an oracle that computes $f(S)$ for given $S \subseteq V$; we consider the running time of this oracle to be $O(1)$.

The following fact is folklore. We provide a proof in the full version (see the supplementary material).

**Fact 2.1** *Let $f$ be a non-negative submodular function and $X_1, X_2, ..., X_k \subseteq X$ be disjoint subsets of $X$. Then*

$$\sum_{i=1}^{k} f(X \setminus X_i) \geq (k - 1)f(X).$$

**Matroids.** A *matroid* is a set family $\mathcal{I} \subseteq 2^V$ with the properties:

- *Downward-closedness*: if $X \subseteq Y$ and $Y \in \mathcal{I}$, then $X \in \mathcal{I}$;
- *Augmentation*: if $X, Y \in \mathcal{I}$ and $|X| < |Y|$, then there exists $e \in Y$ with $X + e \in \mathcal{I}$.

We abbreviate $X \cup \{e\}$ as $X + e$ and $X \setminus \{e\}$ as $X - e$. We assume that the matroid is given as an oracle that, for a given $S \subseteq V$, answers whether $S \in \mathcal{I}$; we consider the running time of this oracle to be $O(1)$. We say that a set $S \subseteq V$ is *independent* if $S \in \mathcal{I}$.

**Matroid exchange graph.** Let $\mathcal{I}$ be a matroid on universe $V$ and $Y$, $Z$ be two independent sets.

**Definition 2.2** *We define the exchange graph for $Y$ and $Z$ with respect to $\mathcal{I}$ as the bipartite graph*

$$(Y \setminus Z, Z \setminus Y, \{(y, z) : Y - y + z \in \mathcal{I}\}).$$

**Lemma 2.3 ([Sch03], Corollary 39.12a)** *If $|Y| = |Z|$, then the exchange graph for $Y$ and $Z$ with respect to $\mathcal{I}$ contains a perfect matching.*

## 2.1 Fair Matroid Monotone Submodular Maximization (FMMSM)

The universe $V$ is partitioned into $C$ sets: $V = V_1 \cup V_2 \cup ... \cup V_C$, where $V_c$ denotes elements of color $c$. Every element has exactly one color. The set of colors is denoted by $[C] = \{1, 2, ..., C\}$. For every color $c \in [C]$ we have *fairness bounds*: lower bound $\ell_c$ and upper bound $u_c$.

The set of upper bounds gives rise to a *partition matroid* that we will denote by $\mathcal{U}$. That is,

$$\mathcal{U} = \{S \subseteq V \mid |S \cap V_c| \leq u_c \ \forall c \in [C]\}.$$

It is well-known that such a collection of sets forms a matroid. We will call a set $S \in \mathcal{U}$ *upper-fair*.

If a set satisfies both the lower and the upper bounds, we say that it is *fair*. That is, we define the family of fair sets $\mathcal{C}$ as follows:

$$\mathcal{C} = \{S \subseteq V \mid \ell_c \leq |S \cap V_c| \leq u_c \ \forall c \in [C]\}.$$

The FMMSM problem asks to find a set $S \in \mathcal{I} \cap \mathcal{C}$ (i.e., fair and independent $S$) that maximizes $f(S)$. We use OPT for the optimal value, i.e., $\text{OPT} = \max_{S \in \mathcal{I} \cap \mathcal{C}} f(S)$. We assume that there exists a fair and independent set, i.e., $\mathcal{I} \cap \mathcal{C} \neq \emptyset$. We say that an algorithm is an $\alpha$-approximation if it outputs a set $S$ with $f(S) \geq \alpha \cdot \text{OPT}$.

For any set $S \subseteq V$ we define its *fairness violation* $\text{fav}(S) := \sum_c \max\{|S \cap V_c| - u_c, \ell_c - |S \cap V_c|, 0\}$. Note that if $S$ is upper-fair, then $\text{fav}(S) = \sum_c \max\{\ell_c - |S \cap V_c|, 0\}$.

**Lemma 2.4 ([EFNTT23], Appendix C)** *There is an exact polynomial-time algorithm for FMMSM for the case when $f$ is a linear function.*

**Matroid intersection.** Given two matroids and a monotone submodular function $f$ defined on $V$, we can define the problem of maximizing a submodular function subject to a matroid intersection constraint similarly to FMMSM. In particular, if we ignore the lower bounds completely, FMMSM turns into the above matroid intersection problem for matroids $\mathcal{I}$ and $\mathcal{U}$.

**Theorem 2.5 ([CCPV11])** *The greedy algorithm gives a $1/3$-approximation to this problem.*

**Theorem 2.6 ([LSV10])** *For any $\delta > 0$ there is a polynomial-time algorithm that gives a $(0.5 - \delta)$-approximation to this problem.*

## 3 Our algorithm

In this section we describe our algorithms: randomized (Theorem 3.4) and deterministic (Section 3.1). We first need to introduce some notions. The proof of Theorem 3.4 will begin by constructing a maximum-cardinality independent and fair set $P$, which will stay unchanged throughout the execution. We also construct an independent and upper-fair set $Y$ of high $f$-value. We will use $P$ as a source of fairness and iteratively trade off $Y$'s value for $P$'s elements in colors that are undersaturated by $Y$.

**Definition 3.1** *Given $Y$ and $P$ as above, we say that a color $c \in [C]$ is undersaturated if $|Y \cap V_c| < |P \cap V_c|$, and oversaturated if $|Y \cap V_c| > |P \cap V_c|$.*

The technical crux of the proof of Theorem 3.4 is Lemma 3.3, in which we show the existence of many disjoint structures, each of which can be used to advance our fairness objective.

**Definition 3.2** *Let $Y$ be an independent and upper-fair set, and let $X \subseteq V$. Define the result $Y'$ of applying $X$ to $Y$ as the symmetric difference $Y' = Y \triangle X$. We say that $X$ is* alternating *(with respect to $Y$) if $Y'$ is independent ($Y' \in \mathcal{I}$) and there is exactly one undersaturated color $c' \in [C]$ and one oversaturated color $c'' \in [C]$ such that for all $c \in [C]$,*

$$|Y' \cap V_c| = |Y \cap V_c| + \begin{cases} 1 & \text{for } c = c', \\ -1 & \text{for } c = c'', \\ 0 & \text{for } c \neq c', c''. \end{cases}$$

*We say that $X$ is* augmenting *if all the above conditions are satisfied, except that there is no color $c''$. In both cases, we say that $X$ increases $c'$.*

Note that we have $|Y'| = |Y|$ if $X$ is alternating and $|Y'| = |Y| + 1$ if $X$ is augmenting. Also, $Y'$ is upper-fair, since the only color where it has more elements than $Y$ is $c'$, and we have $|Y' \cap V_{c'}| = |Y \cap V_{c'}| + 1 < |P \cap V_{c'}| + 1$ (and $P$ is fair).

**Lemma 3.3** *Let $Y$ and $P$ be two independent and upper-fair sets with $|Y| \leq |P|$. Denote*

$$k = \sum_{c \in [C]} \max(0, |P \cap V_c| - |Y \cap V_c|).$$

*Then we may find in polynomial time a collection $X_1, ..., X_k$ of disjoint subsets of $Y \cup P$, of which at least $|P| - |Y|$ many are augmenting and the rest are alternating. Moreover, for every undersaturated color $c$, exactly $|P \cap V_c| - |Y \cap V_c|$ many of the paths increase $c$.*

The full proof can be found in the full version (see the supplementary material).

*Proof sketch.* We consider the so-called (directed) matroid intersection exchange graph for $Y$ and $P$ with respect to $\mathcal{I}$ and $\mathcal{U}$. Inside this graph we carefully construct a subgraph consisting of two matchings $M_{\leftarrow}$ and $M_{\rightarrow}$. $M_{\leftarrow}$ is obtained by invoking Lemma 2.3 on a subgraph, whereas we construct $M_{\rightarrow}$ manually by matching elements of the same colors between $Y$ and $P$. We then algorithmically construct the $k$ paths between appropriately defined sets of sources and sinks in $M_{\leftarrow} \cup M_{\rightarrow}$. Next, we carry out an asymmetric shortcutting process, whose aim is to make sure that the new solution will be independent in the matroid, but also not disrupt the color structure. This allows us to prove that the paths following this postprocessing satisfy Definition 3.2. $\square$

**Theorem 3.4** *There is a randomized polynomial-time algorithm for FMMSM parametrized by $\varepsilon \in (0, 1)$ that outputs a set $S \in \mathcal{I} \cap \mathcal{U}$ (i.e., independent and upper-fair) such that*

- $\mathbb{E}[|S|] \geq (1 - \varepsilon)N$ *with a high-probability tail bound:*
  *for $\delta > 0$, $\mathbb{P}[|S| < (1 - \delta)(1 - \varepsilon)N] \leq \exp(-\Omega_\delta(N))$*

- $\mathbb{E}[f(S)] \geq 0.499 \cdot \varepsilon \cdot \mathrm{OPT}_{\mathrm{MatInt}}$

- *for every $c \in [C]$ we have $\mathbb{E}[|S \cap V_c|] \geq (1 - \varepsilon)\ell_c$*

- *with a high-probability tail bound on the total fairness violation:*
  *for $\delta > 0$, $\mathbb{P}[\mathrm{fav}(S) > (1 + \delta)\varepsilon \sum_c \ell_c] \leq \exp\left(-\Omega_\delta\left(\sum_c \ell_c\right)\right)$*

*where $N$ is the maximum size of a set in $\mathcal{I} \cap \mathcal{U}$, and $\mathrm{OPT}_{\mathrm{MatInt}}$ is the maximum $f$-value of a set in $\mathcal{I} \cap \mathcal{U}$ (clearly we have $\mathrm{OPT}_{\mathrm{MatInt}} \geq \mathrm{OPT}$ as $\mathcal{C} \subseteq \mathcal{U}$).*

We stress that $S$ is upper-fair with probability 1, not only in expectation. We also remark that one can show a similar tail bound for every individual $\ell_c$, though the right-hand side $\exp(-\Omega_\delta(\ell_c))$ may not be meaningful unless $\ell_c$ is large. On the other hand, no such bound can be shown for the $f$-value, which in the worst case can be concentrated on a single element of the universe.

The guarantee $\mathbb{E}[f(S)] \geq 0.499 \cdot \varepsilon \cdot \mathrm{OPT}_{\mathrm{MatInt}}$ of the second bullet point comes from using the local search algorithm of Theorem 2.6 as a subroutine. We can instead use the simpler algorithm of Theorem 2.5 to get a slightly worse guarantee of $\mathbb{E}[f(S)] \geq \frac{1}{3} \cdot \varepsilon \cdot \mathrm{OPT}_{\mathrm{MatInt}}$; we do so in our experimental evaluation.

We give a brief sketch; the full proof can be found in the full version (see the supplementary material).

*Proof sketch of Theorem 3.4.* As the first step, we compute a maximum-cardinality fair and independent set $P$, which may be done in polynomial time by Lemma 2.4. By invoking Lemma 3.3 we can show that $|P| = N$. As the second step, we compute a high-value independent and upper-fair set $Y_0$. Using the algorithm of Theorem 2.6 ([LSV10]) (with $\delta = 10^{-3}$) we get that $f(Y_0) \geq 0.499 \cdot \text{OPT}_{\text{MatInt}}$. We denote

$$k(Y) = \sum_{c \in [C]} \max(0, |P \cap V_c| - |Y \cap V_c|)$$

for any solution $Y$, and $k := k(Y_0)$ to shorten notation. We will perform a number $I$ of iterations which will be $(1 - \varepsilon)k$ in expectation. More precisely, let us set $I = \lceil (1 - \varepsilon)k \rceil$ with probability $(1 - \varepsilon)k - \lfloor (1 - \varepsilon)k \rfloor$, and $\lfloor (1 - \varepsilon)k \rfloor$ otherwise.

We perform $I$ iterations. In the $i$-th iteration, we apply Lemma 3.3 to $Y_{i-1}$ (and $P$) to obtain a collection $X_i^1, ..., X_i^{k(Y_{i-1})}$ of augmenting or alternating sets. We choose one of them, $X_i$, uniformly at random, and apply it to obtain a new solution $Y_i = Y_{i-1} \triangle X_i$. Finally, we return $S := Y_I$.

All solutions $Y_0, ..., Y_I$ are independent and upper-fair. The properties of fairness lower bounds intuitively follow because at every iteration one random fairness violation is removed, and the number of iterations is $\approx (1 - \varepsilon)$ times the initial number of fairness violations $k = k(Y_0)$. Since at least $|P| - |Y_{i-1}|$ of the sets at iteration $i$ are augmenting, the claim about the size of the solution follows similarly. We can also show tail bounds by invoking Chernoff-Hoeffding style bounds for hypergeometric distributions. As for the objective value, we prove that at every step, the expected loss is only a $1/(k - i + 1)$ fraction of the current $f$-value, as we select randomly from among $k - i + 1$ disjoint augmenting or alternating sets. After $I \approx (1 - \varepsilon)k$ iterations we then end up with a telescoping product that simplifies to $\frac{\varepsilon k}{k} f(Y_0)$. □

## 3.1 Deterministic algorithm

Now we turn to our deterministic result, Theorem 1.4. It is powered by a lemma that is an analogue of Lemma 3.3. Their proofs are deferred to the full version (in the supplementary material).

**Lemma 3.5** *For any two matroids $\mathcal{I}_1, \mathcal{I}_2$, let $Y, P \in \mathcal{I}_1 \cap \mathcal{I}_2$ be two sets in their intersection, with $|Y| \leq |P|$. Then we may find in polynomial time a collection $X_1, ..., X_{|P|-|Y|}$ of disjoint subsets of $Y \cup P$ such that for each set $X_i$ we have $Y \triangle X_i \in \mathcal{I}_1 \cap \mathcal{I}_2$ and $|Y \triangle X_i| = |Y| + 1$.*

## 4 Experimental evaluation

We evaluate the performance of our algorithms empirically against prior work and natural baselines closely following the experimental setup of prior work [EMNTT20; EFNTT23], on a suite of benchmarks that are standard in the field: graph coverage, clustering, and recommender systems, under different fairness and matroid constraint settings. Our metrics are the submodular objective value $f(S)$ and total fairness violation $\text{fav}(S)$. All of the considered algorithms return sets that are independent and upper-fair, so the measured fairness violations are all with respect to the lower bounds. LLMs were used to assist in coding. We compare the following algorithms:

- OUR($\varepsilon$) – our algorithm of Theorem 3.4, for a range of settings of $\varepsilon \in \{0.2, 0.5, 0.8\}$. To compute a high-value solution $Y$, we run the natural greedy algorithm, which obtains a $1/3$-approximation (Theorem 2.5), as the local search algorithm of Theorem 2.6 is impractical. The large fair set $P$ is obtained via augmenting paths, ignoring $f$.
- TWOPASS – the algorithm of [EFNTT23] (Theorem 1.1). Since it was originally developed for the streaming setting, to get a fair comparison we simplify away the parts (namely the first pass) whose purpose was ensuring low memory usage. The first step of the algorithm obtains a fair set via augmenting paths (ignoring $f$). This is then divided in two, and each half is extended to an independent and upper-fair solution using a matroid intersection subroutine. For this we employ the greedy algorithm (the original implementation of [EFNTT23] used a swapping algorithm to ensure low memory and linear runtime, but it obtains inferior values).
- LBMI (Lower Bound Matroid Intersection) – an algorithm that always returns a fair set, with no theoretical guarantee on the value. It starts by building a fair set via augmenting paths, ignoring $f$, and then extends to a maximal solution using the greedy algorithm.

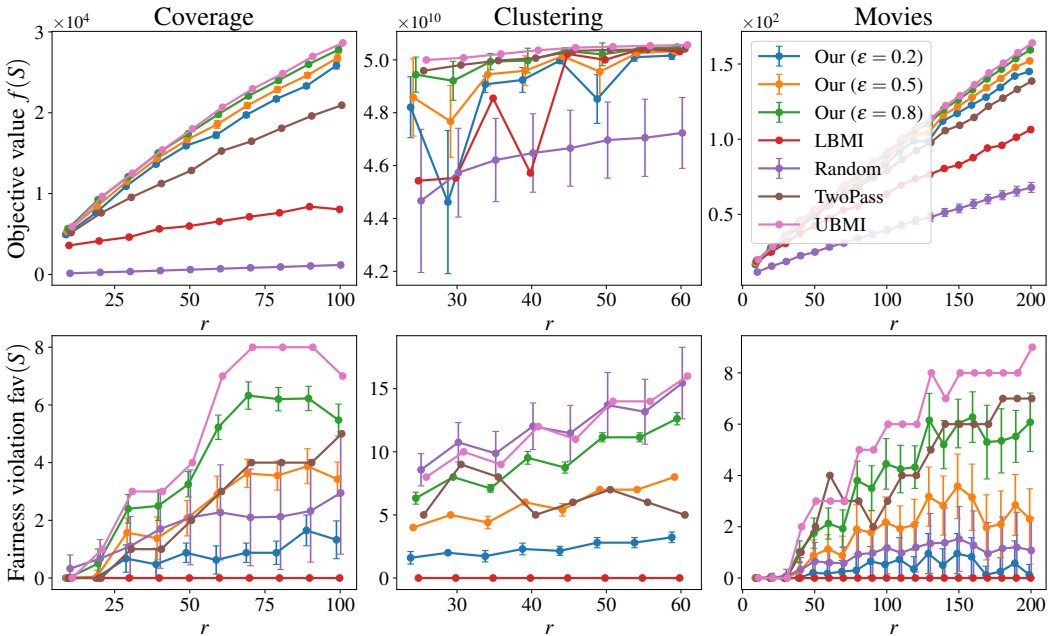

Figure 1: Our experimental results. Each column corresponds to one experiment; the top plot shows the objective value of each algorithm for a range of solution scale factors $r$, and the bottom plot shows fairness violations. For randomized algorithms we report averages, with error bars that correspond to sample standard deviation.

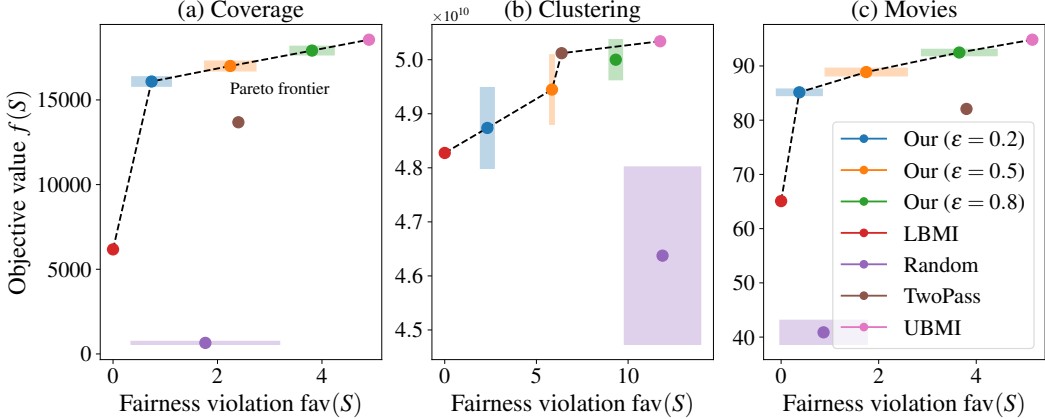

Figure 2: For each experiment and algorithm we take the average objective value and fairness violation over all $r$-values, and plot this as a single point. For randomized algorithms, the colored rectangles correspond to standard deviations. The dashed line corresponds to the Pareto frontier of the trade-off between objective value and fairness violation.

- UBMI (Upper Bound Matroid Intersection) – an algorithm that ignores lower bound constraints and just solves the matroid intersection problem for $\mathcal{I}$ and $\mathcal{U}$ (similar in spirit to MATROID-INTERSECTION from [EFNTT23]). Also here we use the greedy algorithm.

- RANDOM – an algorithm that randomly shuffles the universe and then adds each element if this keeps the solution independent and upper-fair.

For a fair comparison of the main underlying ideas, we made sure that the compared algorithms, particularly OUR and TWOPASS, use the same subroutines for similar tasks; the implementations could likely benefit from heuristically taking $f$ into account rather than ignoring $f$ when building large fair sets, or from some local-search based postprocessing of the final solution. We do not compare to the algorithm of [ETNV24] (Theorem 1.2) as solving the multilinear extension makes it impractical. We repeat the randomized algorithms 40 times. All experiments can be run on commodity hardware (CPU only, single-threaded; we do not report runtimes) and take several hours to finish.

The code for the paper is available at `https://github.com/dj3500/fair-matroid-submodular-neurips2025`.

We outline the experimental scenarios below. In each experiment we vary a solution size scaling factor $r$, which roughly corresponds to the rank of the matroid $\mathcal{I}$.

**Computational complexity.** We start with the complexity of the general randomized algorithm of Theorem 3.4. Firstly, the runtime of constructing $P$ (a maximum-cardinality fair and independent set) via augmenting paths is $O(N^{1.5}|V|)$ (by [Sch03], Chapter 41.2 Notes). To construct $Y$ (an upper-fair and independent set of high $f$-value), we expend $O(N|V|)$ time using the greedy algorithm.

Next, at each of the $I$ iterations, we must (1) recompute the exchange graph between $Y_i$ and $P$, (2) find $M_\leftarrow$ and $M_\rightarrow$ as the subgraph of interest, (3) decompose $M_\leftarrow \cup M_\rightarrow$ into paths, and (4) shortcut these paths. Step (1) takes $O(N^2)$ time, since we query if a directed edge exists between $y$ and $p$ for all $y \in Y$ and $p \in P$. Finding perfect matchings in step (2) takes at most $O(N^3)$ time (in a practical implementation we could use the Hopcroft-Karp algorithm). Decomposing the resulting subgraph into paths takes at most $O(N)$ time. And lastly, shortcutting the paths again takes at most $O(N^2)$ time. Since $I$ can be $\Theta(N)$, the total runtime is at most $O(N^4)$.

A more efficient implementation is possible if $\mathcal{I}$ is a partition matroid. The intersection of two partition matroids can be naturally interpreted as a bipartite multigraph (the colors, i.e., parts of $\mathcal{U}$ are one side, the parts of $\mathcal{I}$ are the other side, and an element corresponds to an edge between the two parts it belongs to). In this case, we may look at the following exchange graph: direct the edges of $Y$ from left to right, and the edges of $P$ from right to left. This directed graph may be decomposed into paths. These paths are *simultaneously feasible*, and so we do not need to recompute an exchange graph at every step (or shortcut). Since there are $O(N)$ edges, the runtime to decompose this directed graph is $O(N)$. Over the $I$ iterations, we have a total runtime of at most $O(N^2)$.

**Graph coverage.** We use the Pokec social network [LK14]. Given a digraph $G = (V, E)$ of users and their friendships, we select a subset $S \subseteq V$ to maximize coverage, defined by $f(S) = \left| \bigcup_{v \in S} N(v) \right|$, where $N(v)$ is the neighborhood of $v$. User profiles include age, gender, height, and weight. We impose a partition matroid on body mass index (BMI). Profiles missing height or weight or with implausible data are removed, yielding a graph with 582,289 nodes and 5,834,695 edges. Users are partitioned into four BMI categories (underweight, normal, overweight, obese), with upper bounds $\lceil \frac{|V_i|}{|V|} r \rceil$ for each group $V_i$. We also enforce fairness by age, with 7 groups: $[1,10], [11,17], [18,25], [26,35], [36,45], [46+]$, no age. We set $\ell_c = \lfloor 0.9 \frac{|V_c|}{|V|} r \rfloor$ and $u_c = \lceil 1.5 \frac{|V_c|}{|V|} r \rceil$. We use $r$ from 10 to 200.

**Exemplar-based clustering.** We use a dataset of 4521 phone calls from a Portuguese bank marketing campaign [MCR14]. The goal is to select a representative subset $S \subseteq V$ for service quality assessment. Each record $e \in V$ is represented as $x_e \in \mathbb{R}^7$ using 7 numeric features, including age and account balance. We impose a partition matroid on account balance, with 5 groups: $(-\infty, 0), [0, 2000), [2000, 4000), [4000, 6000), [6000, \infty)$. Each group $V_i$ has upper bound $r/5$. Fairness is enforced by age, with 6 groups: $[0, 29], [30, 39], [40, 49], [50, 59], [60, 69], [70+]$, and bounds $\ell_c = 0.1r + 2$, $u_c = 0.4r$ for each $c$. We maximize the monotone submodular function [GK10]: $f(S) = \sum_{e' \in V} \left( d(e', 0) - \min_{e \in S \cup \{0\}} d(e', e) \right)$ where $d(e', e) = \|x_{e'} - x_e\|_2^2$ and $x_0$ is the origin. We use $r$ from 30 to 60.

**Recommender system.** We simulate a movie recommendation system using the Movielens 1M dataset [HK16], with about one million ratings for 3900 movies by 6040 users. As in prior work [MB-NTC17; NTMZMS18; EMNTT20; EFNTT23], we compute a low-rank completion of the user-movie matrix [TCSBHTBA01], yielding $w_u \in \mathbb{R}^{20}$ for each user $u$ and $v_m \in \mathbb{R}^{20}$ for each movie $m$. The product $w_u^\top v_m$ estimates user $u$'s rating for movie $m$. For user $u$, the monotone submodular utility for a set $S$ of movies is $f(S) = \alpha \cdot \sum_{m' \in M} \max \left( \max_{m \in S} \left( v_m^\top v_{m'} \right), 0 \right) + (1 - \alpha) \cdot \sum_{m \in S} w_u^\top v_m$, with parameter $\alpha = 0.85$ balancing coverage and personalized user score. We enforce proportional representation of movies by release date using a partition matroid with 9 decade groups (1911–2000), with upper bounds $\lceil 1.2 \frac{|V_d|}{|V|} r \rceil$ for each decade $V_d$. Movies are also partitioned into 18 genres $c$ (colors), with fairness bounds $\ell_c = \lfloor 0.8 \frac{|V_c|}{|V|} r \rfloor$ and $u_c = \lceil 1.4 \frac{|V_c|}{|V|} r \rceil$. We use $r$ from 10 to 200.

### 4.1 Results and discussion

Our results are depicted in Figs. 1 and 2. Similarly as prior work, we observe that enforcing fairness does come at some cost in the utility value, and that the utility values of the algorithms are much better in practice than the theoretical bounds guarantee.

In all three experiments, our algorithms produce solutions whose value is relatively competitive with UBMI, which completely ignores the lower bound constraints and accordingly has the highest fairness violations. In two of the three scenarios (coverage and movies), all OUR algorithms produce a higher $f$-value than all the other baselines (RANDOM, LBMI, and TWOPASS); in particular, TWOPASS is dominated by both OUR(0.2) and OUR(0.5) with respect to both metrics. For clustering the situation is somewhat unclear, but TWOPASS generally does better. In terms of violation of the lower bound fairness constraints, our different settings of $\varepsilon$, as expected, provide a smooth tradeoff. The baseline that guarantees no fairness violations, LBMI, does relatively poorly in terms of $f$-value.

This tunability of $\varepsilon$ is a key strength of our approach, allowing users to select an operating point that best matches their specific requirements for the balance between utility and fairness.

## 5 Conclusion, limitations, broader impact, and future work

In this work we gave an improved algorithm for FMMSM which, for any $\varepsilon > 0$, returns an approximate solution that satisfies an expected $(1 - \varepsilon)$ fraction of each fairness lower bound while satisfying the matroid constraint and the fairness upper bound constraints exactly; the returned solution is also large in size and enjoys high-concentration guarantees. Recent studies have shown that automated algorithms used in decision-making can introduce bias and discrimination. We make progress towards mitigating such effects in problems that can be formulated as submodular maximization under a matroid constraint, which are relevant to a range of applications such as forming representative committees or curating content for news feeds. We show the strong performance of our algorithm empirically on several real-world tasks. As in prior work, we observe a balance between fairness and utility value; however, this "price of fairness" should not be interpreted as fairness leading to inferior outcomes, but rather as a trade-off between two valuable metrics. The parametric nature of our algorithm (the tunable $\varepsilon$ parameter) provides a new tool to help in navigating this balance.

Our work leaves open the exciting question of the approximability of FMMSM (without violations of fairness constraints) and MSPM. Is there a constant-factor approximation algorithm for MSPM? Or is there a superconstant hardness of approximation for FMMSM? (As remarked in [ETNV24], the latter result would give a negative answer to a fundamental question posed by Vondrák [Von13].) We also do not consider non-monotone objective functions or the streaming setting in this work. Finally, it is important to note that the fairness notion we employ, though standard and general, does not capture some notions of fairness considered in the literature (see e.g. [CR18; TWRTZ19]). No universal definition of fairness exists; the choice of which definition to apply is application-dependent and an active area of research.

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

# A   Supplementary Material

We provide a full version of the paper with all omitted content, skipped proofs etc. together with the supplementary material.

