# OpenReview forum: "Improved Algorithms for Fair Matroid Submodular Maximization"
_NeurIPS.cc/2025/Conference — NeurIPS 2025 poster_

### Official Review · Reviewer_AhMU · 2025-07-02

**Clarity:** 3
**Significance:** 3
**Originality:** 3
**Rating:** 5
**Confidence:** 3

**Summary:**

This paper studies the problem of fair matroid monotone submodular maximization (FMMSM); the objective of this problem must take care of fairness constraints in addition to classic matroid optimization. The main results of this paper (Theorems 1.3 and 1.4) have impressive contributions in addition to earlier results, as their algorithms satisfy the fairness constraints to a tighter extent.

**Questions:**

N/A

**Ethical Concerns:**

["NO or VERY MINOR ethics concerns only"]

**Final Justification:**

The paper is strong due to technical novelty and depth. I maintain my score.

**Limitations:**

Yes, in Section 5.

**Quality:**

3

**Strengths And Weaknesses:**

Strengths:
1. This is a rigorous theoretical paper and the theoretical analysis is sound. It is also well-written in a clear, straight-forward style.
2. The improvements on tightness are impressive. It never violates the upper bound, and the lower bound violation factor of $1-\varepsilon$ is substantial, both for the randomized problem (Theorem 1.3) and deterministic solution (Theorem 1.4) with an additional constraint.
3. The techniques (pages 3-4) are novel to the best of my knowledge, and discussed intuitively and clearly in the section.

Weaknesses:
N/A. Although the tightness is still not optimal and constraints make the solution not universally applicable, any concrete progress on this difficult problem is meaningful.

---

> ### Author Rebuttal · Authors · 2025-07-30
>
> Thank you for your thoughtful review.

---

### Official Review · Reviewer_mGZA · 2025-07-03

**Clarity:** 2
**Significance:** 3
**Originality:** 3
**Rating:** 3
**Confidence:** 3

**Summary:**

The paper studies offline monotone submodular maximization subject to a matroid and a fairness constraint, described as follows:

There is a ground set $V$ partitioned into subsets $V_1, V_2, \cdots, V_C$.

For each $c \in [C]$, there are associated lower and upper bounds $l_c$ and $u_c$, respectively.

We are given oracle access to a monotone submodular function $f$ and a matroid $\mathcal{I}$ defined on the ground set $V$.

The objective is to find a set $S$ that maximizes $f(S)$ subject to satisfying the following properties:

$S$ is independent in the matroid ($S \in \mathcal{I}$).

For each  $c \in [C]$: $l_c \le |S \cap V_c| \le u_c$.



In this paper, the authors develop an algorithm that, for each $\epsilon \in (0, 1)$, outputs a solution that satisfies the matroid and upper bound constraints; for each $c \in [C]$ satisfies a relaxed version of its lower bound constraint ($|S \cap V_c| \ge (1 - \epsilon)l_c$) in expectation; and in expectation obtains a submodular value at least $0.499 \cdot \epsilon \cdot OPT$.

**Questions:**

.

**Ethical Concerns:**

["NO or VERY MINOR ethics concerns only"]

**Final Justification:**

Regarding approximation ratio:

My main concern with the result, the weakness of its approximation guarantee, remains a serious issue.

To further clarify this weakness, it should be noted that in the presence of a multiplicative factor of $\epsilon$, the term $0.499$ is just a constant that bears no significant meaning.
I even believe that expressing the guarantee as $0.499 \cdot \epsilon \cdot OPT$, even if correct (with the more accurate form likely being $0.499 \cdot O(\epsilon) \cdot OPT$ according to proof of theorem 3.4), is misleading. It is more transparent to state that the result provides an $O(\epsilon)$-approximation guarantee.

Concerning the remark about the stronger notion of OPT, I understand your remark and agree with the underlying point you intended to convey. However, the issue remains that, while this remark explains the weakness of your guarantee, it does not justify it, since comparing against a stronger notion of OPT was your own choice.
Additionally, regarding your last sentence, I disagree that comparison with a stronger OPT is a plus in your case. Such a comparison is deemed strong as it allows one to argue that the actual ratio is even better than the one stated.
This argument could be meaningful and showcase the strength of the algorithm when the provided approximation ratio is already reasonably strong and acceptable, but this is not the case when the provided lower bound is weak.

___

Regarding rest of the comments:

Although I am not fully convinced by the space constraint explanation (given the no-limit policy on supplementary material), I appreciate the willingness to incorporate the suggested changes. However, as noted earlier, some of the important parts of the paper are currently missing. It is difficult to follow and verify the ideas of the algorithm and its analysis
in the current form of the paper, which warrants substantial revisions. Therefore, I believe the paper requires an additional round of reviews after the updates are made.

**Limitations:**

yes.

**Paper Formatting Concerns:**

I did not notice any major formatting issue.

**Quality:**

2

**Strengths And Weaknesses:**

Strengths:

Clarity and completeness in definitions and problem formulation.

The introduction is informative, it includes insightful discussions about the difficult special case of MSPM, clearly outlines existing results, their limitations and their contribution in comparison.

Although, I could not fully read and verify the correctness of the proofs provided in the supplementary material, they seemed detailed.

Weaknesses:

The approximation ratio of $0.499 \cdot \epsilon$ is quite weak.
The fact that authors compare their solution with a stronger notion of OPT, get a weak approximation guarantee and then blame the weakness of their result on this choice is not justified for me.

The algorithms are not clearly outlined.

The running time of the algorithm is not mentioned in the paper. [I only noticed a computational complexity paragraph in the experimental section(why?) in the supplementary material.

It would have been better if you also included your final results when using the greedy algorithm instead of the local search one. (Right now, the algorithm used in experimental section mismatches the one used for theoretical results.)

---

> ### Author Rebuttal · Authors · 2025-07-30
>
> Thank you for your thoughtful review and your comments, which will help us improve the final version of the paper.
>
> > The approximation ratio of 0.499 eps is quite weak. The fact that authors compare their solution with a stronger notion of OPT, get a weak approximation guarantee and then blame the weakness of their result on this choice is not justified for me.
>
> It is an exciting open problem to obtain stronger approximation guarantees. The main aim of our remark is to point out that in order to do so, one needs to find a way to compare to a more precise bound on OPT. (Though it is also a plus that our guarantee is with respect to a stronger OPT.)
>
> > The algorithms are not clearly outlined.
> > The running time of the algorithm is not mentioned in the paper. [I only noticed a computational complexity paragraph in the experimental section(why?) in the supplementary material.
> > It would have been better if you also included your final results when using the greedy algorithm instead of the local search one. (Right now, the algorithm used in experimental section mismatches the one used for theoretical results.)
>
> Most of the above was due to space constraints; we will implement all of your feedback in the final version of the paper. In particular, we will add pseudocode for the main algorithm, and a version of our results with both Greedy and local search (with Greedy, one gets $1/3$ in place of $0.499$). We do not mention running time in the main body of the paper primarily due to space constraints and because the algorithm clearly runs in polynomial time; these considerations become more important when the implementations are concerned.

---

> > ### Comment · Reviewer_mGZA · 2025-08-08
> > **Response to the rebuttal**
> >
> > Regarding approximation ratio:
> >
> > My main concern with the result, the weakness of its approximation guarantee, remains a serious issue.
> >
> > To further clarify this weakness, it should be noted that in the presence of a multiplicative factor of $\epsilon$, the term $0.499$ is just a constant that bears no significant meaning.
> > I even believe that expressing the guarantee as $0.499 \cdot \epsilon \cdot OPT$, even if correct (with the more accurate form likely being $0.499 \cdot O(\epsilon) \cdot OPT$ according to proof of theorem 3.4), is misleading. It is more transparent to state that the result provides an $O(\epsilon)$-approximation guarantee.
> >
> > Concerning the remark about the stronger notion of OPT, I understand your remark and agree with the underlying point you intended to convey. However, the issue remains that, while this remark explains the weakness of your guarantee, it does not justify it, since comparing against a stronger notion of OPT was your own choice.
> > Additionally, regarding your last sentence, I disagree that comparison with a stronger OPT is a plus in your case. Such a comparison is deemed strong as it allows one to argue that the actual ratio is even better than the one stated.
> > This argument could be meaningful and showcase the strength of the algorithm when the provided approximation ratio is already reasonably strong and acceptable, but this is not the case when the provided lower bound is weak.
> >
> > ___
> >
> > Regarding rest of the comments:
> >
> > Although I am not fully convinced by the space constraint explanation (given the no-limit policy on supplementary material), I appreciate the willingness to incorporate the suggested changes. However, as noted earlier, some of the important parts of the paper are currently missing. It is difficult to follow and verify the ideas of the algorithm and its analysis
> > in the current form of the paper, which warrants substantial revisions. Therefore, I believe the paper requires an additional round of reviews after the updates are made.

---

> > > ### Author Response · Authors · 2025-08-08
> > >
> > > Thank you for your comment.
> > >
> > > > My main concern with the result, the weakness of its approximation guarantee, remains a serious issue. To further clarify this weakness, it should be noted that in the presence of a multiplicative factor of $\epsilon$, the term $0.499$ is just a constant that bears no significant meaning.
> > >
> > > We disagree. Firstly, the algorithm will be instantiated for some value of $\epsilon$, which will make its approximation ratio guarantee concrete. This enables the study of the fairness violation vs approximation tradeoff. For example, a natural consideration is to set $\epsilon=0.5$ and observe that we recover the approximation guarantee of TwoPass [EFNTT23], therefore subsuming that result.
> > >
> > > Secondly, it is indeed natural to desire a result such as: for any $\epsilon>0$, there is an algorithm with a constant approximation ratio that does not depend on $\epsilon$. However, notice that such a result would imply a full resolution of FMMSM (by setting $\epsilon=1/n$), i.e., a constant-factor approximation algorithm without fairness violations; however, many experts believe that the problem is hard to approximate. Therefore, as long as the result does not accomplish this, the dependency on $\epsilon$ will need to surface somehow.
> > > It is an exciting open problem whether one can get, say, a $O(\sqrt{\epsilon})$ dependency, or perhaps an absolute constant but with $n^{O(1/\epsilon)}$ running time. However, we believe the existence of such follow-up questions does not detract from the value of the result provided in this paper.
> > >
> > > > expressing the guarantee as $0.499 \cdot \epsilon \cdot OPT$, even if correct (with the more accurate form likely being $0.499 \cdot O(\epsilon) \cdot OPT$ according to proof of theorem 3.4)
> > >
> > > The guarantee as expressed in the paper is correct and fully supported by the given proofs. There is no $O()$ involved. We strongly disagree with the "even if correct" phrasing, given that no concern with correctness has been raised by any reviewers so far and we believe in the correctness of our proof as shown in the paper. Your concern about correctness (which is just brought up on the last day of the discussion period) does not clearly point out an issue and just refers to the "proof of theorem 3.4".
> > >
> > > > I even believe that expressing the guarantee as $0.499 \cdot \epsilon \cdot OPT$ is misleading. It is more transparent to state that the result provides an $O(\epsilon)$-approximation guarantee.
> > >
> > > While we disagree that explicitly stating a constant can be more misleading than using O-notation to hide that constant, we are happy to state it as $O(\epsilon)$.
> > >
> > > > I appreciate the willingness to incorporate the suggested changes. However, as noted earlier, some of the important parts of the paper are currently missing.
> > >
> > > Again we respectfully disagree. The changes that we will make are minor:
> > > * we will add an analogue of Theorem 3.4 in the experimental section, whose only difference from Theorem 3.4 will be the constant $1/3$ instead of $0.499$ in the approximation guarantee (in the proof we just plug in Greedy, using Theorem 2.5, instead of local search using Theorem 2.6, to compute the set $Y_0$; nothing else changes)
> > > * we will move the already-existing running time analysis to the main body (given the extra 1 page of space); we again stress that this is not needed to establish the main result, because our claim in Theorem 3.4 is "polynomial-time", and it is obvious that our algorithm runs in polynomial time
> > > * we will add pseudocode for the algorithms
> > >
> > > > It is difficult to follow and verify the ideas of the algorithm and its analysis in the current form of the paper, which warrants substantial revisions.
> > >
> > > It is somewhat difficult to argue with this subjective opinion, but outside of the algorithm not being clearly outlined in the form of pseudocode, it is really not clear to us what your concerns here are, or what "substantial revision" you are proposing. (In fact, we would argue that our writing focuses on introducing and explaining the ideas intuitively and bottom-up, rather than relying on pseudocode to state the algorithm, but we will add one for completeness and as a reference.) Readability is very important to us, so please let us know if a particular proof or line can be improved.

---

### Official Review · Reviewer_46DW · 2025-07-03

**Clarity:** 3
**Significance:** 4
**Originality:** 2
**Rating:** 5
**Confidence:** 4

**Summary:**

The authors study the problem of maximizing a submodular function due to two types of constraints, both forming a matroid. The idea is that one type of constraints is standard, due to some structural requirements of the outcome. The second type of constraints comes from the fairness requirements: we want each protected group to be fairly represented in the selected set of elements. The problem has been already studied, and so far an algorithm was known that can violate the fairness constraints by a factor of 2, at the same time obtaining a half of the standard approximation for maximinzing a submodular function. The authors introduce a family of algorithms, parameterised by \epsilon, which trade objective function for the extent to which fairness constraints are satisfied.

**Questions:**

* Does it make sense to combine your algorithm with the two-pass, and always pick the solution that achieves better objective?
* How to set the value of \epsilon in practice?
* Since two-pass and your algorithm use different ideas, is it possible to combine the two, and get an even better solution?

**Ethical Concerns:**

["NO or VERY MINOR ethics concerns only"]

**Final Justification:**

Thank you for your your answers!

**Limitations:**

yes

**Paper Formatting Concerns:**

No concerns.

**Quality:**

3

**Strengths And Weaknesses:**

Strengths:
* this is an important and well motivated problem,
* the algorithm outperforms the currently know  solution with respect to both criterion on some datasets.

Weaknesses:
* the originality is limited since some well-performing algorithm for the exact problem has been already known,
* while it is good to have a spectrum of algorithms that trade objective function for fairness constraints, it is still unclear how \epsilon should be set.

---

> ### Author Rebuttal · Authors · 2025-07-30
>
> Thank you for your thoughtful review.
>
>
> > the originality is limited since some well-performing algorithm for the exact problem has been already known,
>
> While there have indeed been prior attempts at the problem, we would like to stress that none of the prior approaches are able to satisfy more than 1/2 of the fairness lower bound constraint mass, and especially in the crucial special case of bipartite perfect matchings (MSPM), do not guarantee anything for a given vertex (i.e., some unlucky group may never be represented in the solution). In scenarios involving fairness, it is crucial to violate these constraints as little as possible, and this is the focus of our work.
>
>
> > while it is good to have a spectrum of algorithms that trade objective function for fairness constraints, it is still unclear how \epsilon should be set. How to set the value of \epsilon in practice?
>
> In general, a larger epsilon is better if the user considers the objective value more important than the fairness constraints, and a smaller epsilon is better in the opposite case. How exactly to set it is up to the user and will be application-dependent.
>
> We remark that a single execution of the algorithm can return solutions for all possible epsilon values (as we iteratively apply augmenting or alternating sets, this corresponds to epsilon decreasing). So a user could also, for example, adaptively keep running the algorithm as long as the objective value remains acceptable, and obtain better and better fairness (without paying any extra factor in the running time).
>
>
> > Does it make sense to combine your algorithm with the two-pass, and always pick the solution that achieves better objective? Since two-pass and your algorithm use different ideas, is it possible to combine the two, and get an even better solution?
>
> It is not clear to us how to combine the ideas of the two algorithms. In practice, one can run the two and select a “better” solution, though care must be taken as to how this is selected so that new bias is not introduced (e.g. if we select the higher objective value, this could risk losing the fairness guarantees).

---

### Official Review · Reviewer_6JcZ · 2025-07-05

**Clarity:** 3
**Significance:** 2
**Originality:** 2
**Rating:** 3
**Confidence:** 3

**Summary:**

This paper presents an improved algorithm for the Fair Matroid Max-Sum Maximization (FMMSM) problem. For any \epsilon> 0, the proposed method returns an approximate solution that satisfies the matroid constraint and all fairness upper bound constraints, while ensuring that each fairness lower bound is met in expectation up to a (1 − \epsilon) factor. The paper is really about constrained submodular maximization with complicated constraints, such as partition matroid and lower fairness constraints.

**Questions:**

I would really like to see a real application of this setup.

**Ethical Concerns:**

["NO or VERY MINOR ethics concerns only"]

**Limitations:**

Even though I am not an expert in the fairness literature, I am less convinced that this formulation is the right one.

**Quality:**

3

**Strengths And Weaknesses:**

While the theoretical analysis is technically sound, the contribution feels fairly incremental given the existing literature, particularly [EFNTT23] and [ETNV24], which already address many of the core ideas and challenges in this setting. Additionally, I find the problem formulation difficult to justify, I am not really sure why why this setup captures a meaningful or realistic fairness scenario, and it comes across as somewhat ad hoc without a compelling motivating application.

The experimental results in Section 4, though easy to follow, also feel artificially constructed. The examples, such as exemplar-based clustering and graph coverage, are nearly identical to those commonly seen in the submodular maximization literature. This raises concerns about how well the proposed method generalizes beyond these stylized benchmarks. A broader or more practical evaluation would help establish the real-world relevance of the approach.

---

> ### Author Rebuttal · Authors · 2025-07-30
>
> Thank you for your thoughtful review.
>
>
> > While the theoretical analysis is technically sound, the contribution feels fairly incremental given the existing literature, particularly [EFNTT23] and [ETNV24], which already address many of the core ideas and challenges in this setting.
>
> While there have indeed been prior attempts at the problem, we would like to stress that none of the prior approaches are able to satisfy more than 1/2 of the fairness lower bound constraint mass, and especially in the crucial special case of bipartite perfect matchings (MSPM), do not guarantee anything for a given vertex (i.e., some unlucky group may never be represented in the solution). In scenarios involving fairness, it is crucial to violate these constraints as little as possible, and this is the focus of our work.
>
>
> > Additionally, I find the problem formulation difficult to justify, I am not really sure why why this setup captures a meaningful or realistic fairness scenario, and it comes across as somewhat ad hoc without a compelling motivating application. (...) Even though I am not an expert in the fairness literature, I am less convinced that this formulation is the right one.
>
> The problem formulation and the group fairness model are standard in the field, for both submodular maximization [CHV18; EMNTT20; WFM21; TY23; TYM23; YT23; EFNTT23; ETNV24] as well as a multitude of other optimization problems, such as clustering [CKLV17, KAM19, JNN20, HMV23], voting [CHV18], data summarization [CKSDKV18], ranking [CSV18] or matching [CKLV19]. It is important to remember that no universal definition of fairness exists, and the choice of which definition to use is application-dependent and an active area of research. However, our notion of fairness is standard, general, well-studied for many problems, and it also generalizes several other fairness notions, such as proportional representation [Mon95; BLS17], diversity constraints [CCRL13; Bid06], or statistical parity [DHPRZ12].
>
> References not included in the submission:
>
> * [KAM19] Matthäus Kleindessner, Pranjal Awasthi, Jamie Morgenstern. Fair k-Center Clustering for Data Summarization. Proceedings of the 36th International Conference on Machine Learning (ICML), PMLR 97:3448-3457, 2019.
>
> * [JNN20] M. Jones, H. Nguyen, and T. Nguyen. Fair k-centers via maximum matching. In Proceedings of the International Conference on Machine Learning (ICML), pages 4940–4949, 2020.
>
> * [HMV23] Sedjro Salomon Hotegni, Sepideh Mahabadi, Ali Vakilian. Approximation Algorithms for Fair Range Clustering. Proceedings of the 40th International Conference on Machine Learning (ICML), PMLR 202:13270-13284, 2023.
>
> * [TY23] Shaojie Tang and Jing Yuan. Beyond submodularity: a unified framework of randomized set selection with group fairness constraints. Journal of Combinatorial Optimization, 45(4):102, 2023.
>
> * [TYM23] Shaojie Tang, Jing Yuan, and Twumasi Mensah-Boateng. Achieving long-term fairness in submodular maximization through randomization. arXiv preprint arXiv:2304.04700, 2023.
>
> * [YT23] Jing Yuan and Shaojie Tang. Group fairness in non-monotone submodular maximization. Journal of Combinatorial Optimization, 45(3):88, 2023.
>
> * [WFM21] Yanhao Wang, Francesco Fabbri, and Michael Mathioudakis. Fair and representative subset selection from data streams. In Proceedings of the Web Conference 2021, pages 1340–1350, 2021.
>
> > The experimental results in Section 4, though easy to follow, also feel artificially constructed. The examples, such as exemplar-based clustering and graph coverage, are nearly identical to those commonly seen in the submodular maximization literature. This raises concerns about how well the proposed method generalizes beyond these stylized benchmarks. A broader or more practical evaluation would help establish the real-world relevance of the approach.
>
> Also here, we see it as an asset of our paper that we conform with the standard benchmarks and evaluations in the field. The focus is to evaluate the effectiveness of our proposed algorithm and compare it fairly against prior approaches, rather than to find new applications of the well-established problem setting.

---

> > ### Author Response · Authors · 2025-08-07
> >
> > Hi,
> >
> > Given that we are approaching the end of the discussion period, we are reaching out to ask if we have addressed your concerns about the problem formulation.
> >
> > Thank you

---

### Official Review · Reviewer_QVWu · 2025-07-08

**Clarity:** 3
**Significance:** 3
**Originality:** 3
**Rating:** 5
**Confidence:** 3

**Summary:**

This paper considers fair submodular maximization subject to a matroid constraint. This is much more challenging than the simpler cardinality constraint version of the problem, where it has previously been shown that the fairness constraint plus cardinality constraint are equivalent to an instance of submodular maximization with a single matroid constraint. This problem has previously been considered before by [EFNTT23,ETNV24], but an issue with both of these results is that the proposed algorithms do not actually guarantee to meet the bottom fairness constraint (they either only meet them in expectation or by a constant factor).

**Questions:**

- Is it possible to use concentration bounds and the algorithm of ETNV24 in order to meet the lower bounds with high probability? If so, what is the advantage of your algorithm in terms of meeting the fairness constraints?
- Related to above question, could you describe the theoretical guarantees of your algorithm compared to ETNV24 in more depth?
- What do you mean by that you are "giving guarantees for every individual group rather than only in aggregate" in contrast to related work?

**Ethical Concerns:**

["NO or VERY MINOR ethics concerns only"]

**Final Justification:**

Update: After reading the rebuttal and other reviews, I have raised my score to an accept. My primary issue was the theoretical relationship with ETNV24, and the rebuttal of the authors has addressed this.

**Limitations:**

Yes

**Quality:**

3

**Strengths And Weaknesses:**

Strengths
- The problem is both interesting and challenging.
- Algorithms and theoretical techniques appear novel to me.
- Experimental results are included.

Weaknesses
- I am confused about how the results of this work improve on ETNV24, which from my understanding achieves the bounds on fairness in expectation. In particular, I have listed two questions below that would help clarify the contributions of this work relative to existing work. If this relationship can be better clairified I am willing to increase my score.
- The experiments do not compare to ETNV24 since they say using the multilinear extension is impractical, but this comparison seems the most important from my perspective since it is the algorithm in the classic offline setting.

---

> ### Author Rebuttal · Authors · 2025-07-30
>
> > I am confused about how the results of this work improve on ETNV24, which from my understanding achieves the bounds on fairness in expectation. In particular, I have listed two questions below that would help clarify the contributions of this work relative to existing work. If this relationship can be better clairified I am willing to increase my score.
>
> Happy to clarify; see answers below. Thank you for your thoughtful review and these questions, which will help us improve the writeup for the final version.
>
> > The experiments do not compare to ETNV24 since they say using the multilinear extension is impractical, but this comparison seems the most important from my perspective since it is the algorithm in the classic offline setting.
>
> ## Practicality of [ETNV24]:
>
> The algorithm of [ETNV24] (Theorem 3.3 in that paper) involves first approximately solving the multilinear relaxation (and then applying randomized swap rounding to the approximate fractional solution). This first step is done using the Continuous Greedy algorithm of [CCPV11], which has a very high polynomial running time and is widely regarded as impractical, see below:
> * [CCPV11]: “Our algorithm is conceptually quite simple and easy to implement; however, its running time is a different question. The variant described here would run in time on the order of $O(n^8)$, using $O(n^7)$ oracle queries”
>
>
> * [HKMS19]: “In particular, it is known that the Continuous Greedy algorithm achieves the tight $(1 − 1/e)$ approximation guarantee (...) with a prohibitive query complexity of $O(n^8)$”
>
>
> * [SM25]: “algorithms that … solve a continuous relaxation of the problem based on the multilinear relaxation, which is known to be impractical [5, 9, 15]. These algorithms do not scale to larger problem instances”
>
>
> References:
>
> * [CCPV11] Gruia Calinescu, Chandra Chekuri, Martin Pál, Jan Vondrák. Maximizing a Monotone Submodular Function Subject to a Matroid Constraint (SIAM Journal on Computing 2011)
>
> * [ETNV24] Marwa El Halabi, Jakub Tarnawski, Ashkan Norouzi-Fard, Thuy-Duong Vuong. Fairness in Submodular Maximization over a Matroid Constraint (AISTATS 2024)
>
> * [HKMS19] Amin Karbasi, Hamed Hassani, Aryan Mokhtari, Zebang Shen. Stochastic Continuous Greedy ++: When Upper and Lower Bounds Match (NeurIPS 2019)
>
> * [SM25] Fabian Spaeh, Atsushi Miyauchi. An asymptotically optimal approximation algorithm for multiobjective submodular maximization at scale (ICML 2025)
>
>
> > Is it possible to use concentration bounds and the algorithm of ETNV24 in order to meet the lower bounds with high probability? If so, what is the advantage of your algorithm in terms of meeting the fairness constraints? Related to above question, could you describe the theoretical guarantees of your algorithm compared to ETNV24 in more depth?
>
> Let us stress that an algorithm achieving both lower and upper fairness bounds only in expectation is quite a weak guarantee. For clarity let us focus on the MSPM (the bipartite perfect matching) case. There, such an algorithm actually gives no theoretical guarantee. In principle, it could return a solution that, for a vertex $v$, has 0 incident edges with probability 0.99, and 100 incident edges with probability 0.01, and still meet the in-expectation guarantee. If one wants to return a matching (i.e. if the upper bound constraints are hard, as they are for all the other algorithms considered in our paper), one has to then delete 99 of the 100 edges, which will spoil both the objective value and the solution cardinality.
>
> The [ENTV24] algorithm does satisfy concentration bounds (see Theorem 3.3 there, as well as a slightly better bound in Remark B.1), but as stated, these are only meaningful for large values of $\ell_c$. By opening up the algorithm and its proof, one can use the bounds for randomized swap rounding [CVZ10, Theorem II.3] to get some meaningful bound: $P(|S \cap \delta(v)| \ge 1) \ge 1 - e^{-1/2} \approx 0.4$. This means that, for a bipartite graph with $n+n=2n$ vertices, $0.4n$ right-side vertices will be guaranteed to have an incident edge in $S$; when we select one edge per right-side vertex, we can expect a matching of size $0.4n$, and one can perhaps hope for a similar bound on submodular value, i.e., $0.4 (1-1/e) OPT \approx 0.25 OPT$. But a simple greedy algorithm gets $0.5 n$ and $OPT/3$ respectively. This is why on line 78 we say that for MSPM, [ENTV24] gives no improvement upon the simple greedy algorithm; we will expand this in the final version.
>
> It is not clear how one could use concentration bounds to satisfy lower bounds with high probability.
>
> Let us remark that the limitations of the [ENTV24] algorithm are intuitively to be expected, as it is based on the multilinear relaxation, and as such is subject to the $\Omega(\sqrt{|E_G|})$ integrality gap proved also in [ENTV24].
>
> Reference:
> * [CVZ10] Chandra Chekuri; Jan Vondrak; Rico Zenklusen. Dependent Randomized Rounding via Exchange Properties of Combinatorial Structures (FOCS 2010)
>
> > What do you mean by that you are "giving guarantees for every individual group rather than only in aggregate" in contrast to related work?
>
> Apologies for the confusion; this is in contrast to the simple greedy strategy discussed on lines 75-82. We will improve the writing there.

---

> > ### Author Response · Authors · 2025-08-06
> >
> > Just reaching out to check if we have managed to clarify your concerns? Thank you!

---

### Decision · Program_Chairs · 2025-09-17

**Decision:**

Accept (poster)

**Comment:**

The paper proposes a new algorithm for the matroid submodular optimization problem under matroid constraint, improving previous work on the same problems.

The problem studied in the paper is interesting and clearly improve on previous work.

Few improvements suggested by multiple reviewers are:
- more in depth description of the novelty of the work, currently the results look a bit incremental
- more in depth comparison with ETNV24